# Impact of parental education on number of under five children death per mother in Bangladesh

Farzana Afroz⊕, Md. Muddasir Hossain Akib⊕, Bikash Pal⊕⊕*, Abida Sultana Asha⊕

Department of Statistics, University of Dhaka, Dhaka, Bangladesh

⊕ All the authors contributed equally to this work.
* bikashpal@du.ac.bd

**Data availability statement:** The dataset utilized for this study is accessible in the

## Abstract

One of the leading challenges of social development is the reduction of children's deaths under the age of five. The primary focus of this research is to study the potential impact of parental education on under five children death in Bangladesh utilizing a secondary data-set extracted from the Bangladesh Demographic and Health Survey (BDHS), 2017–18. The total count of deceased children within a family is a non-negative numerical variable. The mean number of under five children death per 100 mothers is found to be 20 with variance of around 27, which indicates the presence of overdispersion. As the response variable exhibits 84.2% zero counts, we have considered three regression models in this research; Poisson model, zero-inflated Poisson model, and zero-inflated negative bino-mial model. Finally, zero-inflated negative binomial model, exhibiting the lowest AIC value, indicates that both maternal and paternal education have significant protective impact on under five children death. Specifically, greater levels of formal education achieved by the parents are associated with a decreased rate of children death.

## Introduction

Child mortality stands as a pivotal measure of a nation's prosperity and societal advance-ment, resonating in the global discourse on public health. The under five child mortality rate, representing the risk of a child dying before reaching the age of five, serves as a critical gauge of socioeconomic development and living standards. The Sustainable Development Goals (SDGs) aim for a significant reduction in neonatal and under-five mortality rates by 2030, targeting 12 and 25 deaths per 1,000 live births, respectively [1]. Currently, 79 nations grapple with under five mortality rates exceeding 25 per 1000 live births [2]. The case of Bangladesh is noteworthy, as under-five mortality rates witnessed a substantial decrease. Specifically, between 1990 and 2018, there was a significant drop in mortality rates, attributed to the effective implementation of the Millennium Development Goals (MDGs) [3]. As Bangladesh strives to align with SDG 3, aiming to reduce mortality for newborns and children under five to 25 per 1000 live births by 2030 [4], a critical examination of the strategies employed becomes imperative.

following DHS repository https://dhsprogram.com/data/available-datasets.cfm.

**Funding:** The author(s) received no specific funding for this work.

This paper scrutinizes parental education as a key determinant of child mortality reduction in Bangladesh. Drawing on nationwide survey data, our study underscores the pivotal role of primary caregivers' education as a potential pathway for Bangladesh to achieve the SDGs. While various studies have explored socioeconomic and demographic factors influencing child mortality in Bangladesh, only a limited number have delved into the specific impact of parental education. Addressing this research gap, our study specifically underscores the impact of paternal education on modelling mortality of children in Bangladesh. Despite the widely held belief in the contribution of both parents' education to child health, existing studies present mixed findings [5–9]. For instance, research in Indonesia suggests that both maternal and paternal education exhibit a similar association with under five mortality [8], while others indicate a weaker or insignificant link between paternal education and child mortality compared to maternal education [5,6,10].

Increasing evidence emphasizes the protective effect of maternal education on child mortality [11,12]. Mothers who have attained advanced education levels typically exhibit greater familiarity with services related to maternal health, leading to improved health outcomes during childbirth and the postnatal period. Moreover, their awareness of advancements in healthcare facilitates access to essential services for both themselves and their newborns. Maternal education also equips mothers with the knowledge to identify health discrepancies in their children early on, fostering timely interventions.

While the effect of paternal education on child mortality remains debated, Akter et al. [13] discovered that higher maternal education is correlated with a 38% reduction in under five mortality, whereas greater paternal education contributed a 16% reduction. In the socio-cultural context of Bangladesh, where fathers are often considered household heads responsible for financial matters, their lack of formal education may hinder their awareness of necessary healthcare for their offspring. This knowledge gap might lead them to opt for cost-effective alternatives, such as traditional village doctors, potentially compromising newborn health.

In line with this backdrop, our research examines the link between the parental education level and the under five mortality rates in Bangladesh. Additionally, we consider other covariates to account for potential confounding effects of parental education. Various regression models have been explored in existing literature to analyze count responses, including the Poisson Regression model (PRM), Negative Binomial Regression model (NBRM), and Generalized Poisson Regression model (GPRM) [14]. The application of PRM is contingent on the assumption that the value of the mean and the variance of the response variable are equal. The GPRM stands out as a versatile tool adept at handling both overdispersion and underdispersion, while the NBRM is specifically recommended for addressing overdispersion [15,16]. However, the presence of excess count of zeros in the response variable necessitates a nuanced approach. In this study, we opt for the Poisson Regression Model, the zero-Inflated Poisson Regression model, and the zero-inflated negative binomial regression [15] to effectively model the number of children's deaths per mother within the regression framework.

Our study seeks to enhance the existing literature by clarifying trends in the relationship between parental education and under five mortality, utilizing national survey data from the Bangladesh Demographic and Health Survey (BDHS), 2017–18. On the policy front, our findings may inform strategies for combating child mortality by shedding light on the evolving significance of parental education.

## Methods

### Data and variables

This study utilizes a dataset sourced from the 2017–18 Bangladesh Demographic and Health Survey (BDHS). This survey, part of a series of national-level population and health surveys, is

designed to provide comprehensive data on the demographic and health conditions of women and children in Bangladesh [17].

This study centers on child mortality rates per mother in Bangladesh, focusing specifically on children under the age of five. The data utilized originates from the BDHS 2017–18, which collected information from 20,127 ever-married women aged 15–49 on 5,243 variables. Among all the individuals, data from women who have never given birth have been considered irrelevant with the objective of the study, and hence discarded. And lastly, the final analysis included 18,134 women who had given birth.

In this paper, the number of children's death per mother (CDPM) has been derived from the variable- age at death (months imputed), where mothers whose children died before the age of five (60 months) have been taken into account and it resulted in a count variable. Therefore, the count response variable of our interest is the number of under five children death for each mother.

As the explanatory variables, a total of nine covariates have been considered here, namely, administrative division (region), type of place of residence, mother's education level, father's education level, wealth index, birth status, exposure to media and mother's working status. As some data on specific variables were not directly available in the 2017–18 BDHS dataset, we adjusted their categorization for analysis. For instance, to define the "exposed to media" variable, we grouped mothers based on their regular engagement with activities such as reading newspapers or magazines, listening to the radio, or watching television on a weekly basis [18]. In the variable birth status, a mother who has given birth to more than two children has been categorized as multiple, while a mother with only one or two children belongs to the category of single/double.

### Ethics approval and consent to participate

The Institutional Review Board of ICF International, Rockville, Maryland, USA (formerly Macro International) reviewed and approved the Demographic and Health Surveys (DHS) Program, including the 2017–18 Bangladesh DHS. Additionally, the 2017–18 Bangladesh DHS received approval from the Bangladesh Medical Research Council. This survey was conducted by the National Institute of Population Research and Training (NIPORT) under the authority of the Government of the People's Republic of Bangladesh, with financial support from USAID/Bangladesh. Informed consent was obtained from each survey participant prior to their involvement, and those who did not consent were not included in the survey.

### Statistical analysis

For analysing under five children death count data, we considered three regression models Poisson regression model, zero-inflated Poisson model and zero-inflated negative binomial model, in the context of generalized modelling framework [19]. The basic Poisson regression model with equidispersion assumption can be used for analysing count data, however in most of the cases the data are overdispersed. When overdispersion is present, negative binomial distribution is preferred which allows Poisson mean to vary following gamma distribution. In data with excess zero count the zeros may not arise by the same mechanism as the other count part [20]. To put up with the extra zeros observed in the data, zero inflated Poisson (ZIP) model is proposed [21]. In zero inflated data the mean and variance may not be equal, to accommodate both zero inflation and overdispersion a natural choice of the model can be the zero inflated negative binomial (ZINB) distribution [22].

In zero-modified Poisson regression model, it is assumed that two different data generation processes exist. Bernoulli trials are utilized to set the probabilities for zero counts $(p_i)$ and for counts derived from the Poisson distribution $(1 - p_i)$. Considering a sample of size $n$, the zero modified model for response $Y_i$, $(i = 1, 2, \ldots, n)$ can be written as

$$P(Y_i = 0) = f_1(0) + (1 - f_1(0)) f_2(0)$$

$$P(Y_i = k) = (1 - f_1(0)) f_2(k), \quad k = 1, 2, 3, \ldots$$

where $f_1(.)$ represents the binary component and $f_2(.)$ denotes the count component of the data. The binary part is typically modeled using a logit model, while the count part is often represented with a Poisson or negative binomial model leading to zero inflated Poisson (ZIP) and zero-inflated negative binomial (ZINB) models respectively. The zero inflated Poisson (ZIP) model can be expressed as

$$P(Y_i = y_i) = \begin{cases} p_i + (1 - p_i) e^{-\mu_i} & \text{if } y_i = 0 \\ (1 - p_i) \dfrac{e^{-\mu_i} \mu_i^{y_i}}{y_i!} & \text{if } y_i > 0 \end{cases},$$

where $\mu_i$ is the mean of the standard Poisson distribution. Mean and variance of ZIP model are $E(Y_i) = (1 - p_i)\mu_i$ and $V(Y_i) = (1 - p_i)(\mu_i + \mu_i^2) - ((1 - p_i)\mu_i)^2$, respectively. Again, the zero-inflated negative binomial (ZINB) model can be expressed as

$$P(Y_i = y_i) = \begin{cases} p_i + (1 - p_i) \left[ \left( \dfrac{\phi}{\mu_i + \phi} \right)^\phi \right] & \text{if } y_i = 0 \\ (1 - p_i) \dfrac{\Gamma(y_i + \phi)}{y_i! \Gamma(\phi)} \left( \dfrac{\mu_i}{\mu_i + \phi} \right)^{y_i} \left( \dfrac{\phi}{\mu_i + \phi} \right)^\phi & \text{if } y_i > 0 \end{cases},$$

where $\phi$ indicates the over dispersion parameter. Mean and variance of ZINB model are $E(Y_i) = (1 - p_i)\mu_i$ and $V(Y_i) = (1 - p_i)\mu_i \left(1 - p_i\mu_i + \mu_i / \phi\right)$, respectively.

This research employed all three models for data analysis, ultimately selecting the model that exhibited the smallest AIC [27]. For a model with $p$ parameters, AIC can be presented as, $\text{AIC} = -2l + 2p$, where $l$ represents the log-likelihood. For the purpose of exploring the effects of the covariates on the count data incidence rate ratio (IRR) is used, which can be written as: $IRR_j = \exp\left(\hat{\beta}_j\right)$, where $\hat{\beta}_j$ represents the estimated regression coefficient for the $j$-th covariate $(j = 1, 2, \ldots, p - 1)$.

## Results

The histogram presented in Fig 1 indicates that the occurrence of more than two child deaths is extremely rare. Additionally, it highlights that a significant majority of mothers (84.21%) experienced no child death. Since the frequency for zero in count of children death per mother is extremely high, we call this a zero inflated count data.

The existence of overdispersion in the dataset becomes apparent, as indicated by Table 1, where the variance of the number of under five children deaths per 100 mothers [27] significantly exceeds the mean number of under-five children deaths per 100 mothers [20].

Therefore, it is easily understood that in addition to the excess zeros in the dataset, overdispersion must also be addressed. Our study employs three regression models; the Poisson Regression Model, zero-Inflated Poisson Regression model, and zero-inflated negative binomial regression. To determine the optimal model fit for our data, our study compares the models using the Akaike Information Criteria (AIC).

**Histogram for the number of under five children death per mother.**

**Fig 1. Histogram for the number of under five children death per mother.**

**Table 1. Descriptive statistics for the number of under five children death per 100 mothers.**

| Characteristics | CDPM |
| --- | --- |
| Size of the Sample (Total number of respondents) | 18134 |
| Sample Mean | 20 |
| Sample Variance | 27 |

As shown in Table 2, the zero-inflated negative binomial exhibits the lowest AIC value (15371.97) among all three models. Thus, we will focus on the resulting values obtained from fitting the zero-inflated negative binomial regression model. Our study reveals that factors such as region, mother's education level, father's education level, mother's age at first birth, wealth index, and birth status significantly affect the number of children's deaths per mother in Bangladesh. The mean number of deaths of children per mother in the Chittagong division is significantly (p-value = 0.041, IRR = 0.851) 14.9% lower than that of the Dhaka division. Urban or rural residency was found to be insignificant. All the categories included in the mother's educational level have been found to be significant. The mean numbers of deaths of children for mothers with primary, secondary, and higher levels of education are respectively 22%, 33.4%, and 39.9% lower in comparison to the number of deaths of children for mothers with no education. Similar outcomes have also been observed regarding the educational levels of fathers. The average numbers of deaths of children per mother are respectively 8.4%, 17.5%, and 20% lower for fathers with primary, secondary, and higher levels of education

**Table 2. Results obtained from different regression models along with their respective AIC values.**

| | Poisson regression | | Zero-Inflated Poisson | | Zero-Inflated negative binomial | |
|---|---|---|---|---|---|---|
| Covariates | IRR | p-value | IRR | p-value | IRR | p-value |
| **Intercept** | 0.065 | **<0.001** | 0.076 | **<0.001** | 0.082 | **<0.001** |
| **Region** | | | | | | |
| Dhaka | – | – | – | – | – | – |
| Barisal | 1.006 | 0.925 | 0.904 | 0.477 | 0.938 | 0.462 |
| Chittagong | 0.916 | 0.200 | 0.815 | **0.099** | 0.851 | **0.041** |
| Khulna | 0.932 | 0.340 | 0.630 | **0.001** | 0.984 | 0.852 |
| Mymensingh | 1.053 | 0.463 | 0.887 | 0.302 | 0.968 | 0.704 |
| Rajshahi | 1.089 | 0.213 | 0.822 | 0.155 | 1.030 | 0.719 |
| Rangpur | 0.970 | 0.667 | 0.784 | **0.054** | 0.987 | 0.881 |
| Sylhet | 1.142 | **0.051** | 1.083 | 0.508 | 1.077 | 0.346 |
| **Place Type** | | | | | | |
| Rural | – | – | – | – | – | – |
| Urban | 1.032 | 0.432 | 1.094 | 0.151 | 0.989 | 0.832 |
| **Respondent's Education Level** | | | | | | |
| No education | – | – | – | – | – | – |
| Primary | 0.764 | **<0.001** | 0.724 | **<0.001** | 0.780 | **<0.001** |
| Secondary | 0.657 | **<0.001** | 0.648 | **<0.001** | 0.666 | **<0.001** |
| Higher | 0.512 | **<0.001** | 0.365 | **<0.001** | 0.601 | **0.001** |
| **Partner's Education Level** | | | | | | |
| No education | – | – | – | – | – | – |
| Primary | 1.001 | 0.969 | 1.076 | 0.330 | 0.916 | **0.098** |
| Secondary | 0.888 | **0.028** | 0.997 | 0.969 | 0.825 | **0.003** |
| Higher | 0.856 | **0.088** | 0.923 | 0.627 | 0.800 | **0.032** |
| **Respondent's Age at 1st Birth** | | | | | | |
| <18 | – | – | – | – | – | – |
| 18–25 | 0.880 | **<0.001** | 0.957 | 0.065 | 0.833 | **<0.001** |
| >25 | 1.037 | 0.798 | 1.500 | **<0.001** | 0.910 | 0.563 |
| **Respondent's Working Status** | | | | | | |
| No | – | – | – | – | – | – |
| Yes | 1.016 | 0.658 | 0.938 | 0.247 | 1.033 | 0.441 |
| **Media Exposure** | | | | | | |
| Not Exposed | – | – | – | – | – | – |
| Exposed | 0.968 | 0.433 | 0.939 | 0.369 | 0.974 | 0.590 |
| **Wealth Index** | | | | | | |
| Poor | – | – | – | – | – | – |
| Middle | 0.979 | 0.626 | 1.099 | 0.213 | 0.999 | 0.986 |
| Rich | 0.845 | **0.015** | 0.842 | 0.103 | 0.869 | **0.078** |
| **Birth Status** | | | | | | |
| Single/double | – | – | – | – | – | – |
| Multiple | 8.348 | **<0.001** | 10.309 | **<0.001** | 7.441 | **<0.001** |
| **AIC** | 15472.00 | | 15402.43 | | 15371.97 | |

in comparison to the number of deaths of children per mother married to fathers with no education. Statistically significant results were obtained for mothers aged 18–25 at their first birth (p-value < 0.001), with the mean number of child deaths per mother being 16.7% lower

than for mothers younger than 18 at their first birth. However, the working status of mothers and their exposure to media did not show significant impacts on the number of child deaths. The wealth index significantly affects the number of child deaths per mother. Women from rich families have a 13.1% lower average compared to those from poor families. Finally, the explanatory variable birth status is statistically significant (p-value < 0.001), and an IRR value of 7.441 indicates that the average number of child deaths per mother with multiple births is 644.1% higher than for those with single or double births.

## Discussion

The inferential findings outlined in this research emphasize the significant impact of parental education on shaping child mortality in Bangladesh. The results suggest a clear association between elevated levels of parental education and a reduction in the average number of child deaths per mother, consistent with earlier research on this subject [5,23]. The discussion emphasizes the multifaceted benefits of maternal education, ranging from increased awareness about health and nutrition to improved decision-making abilities and greater knowledge about disease prevention. These factors collectively contribute to a more favourable environment for child health and well-being. The shift towards educated females participating in the workforce has led to a positive change in societal norms, enabling women to voice their opinions and make informed decisions. Additionally, the research highlights the beneficial influence of education on men, noting that fathers with higher education levels are more likely to support women in making reproductive and healthcare decisions.

While education emerges as a significant factor, the study acknowledges the influence of other sociodemographic factors on child mortality. Maternal age at first birth emerges as a significant contributor, with a consistent trend showing a decrease in the mean number of child deaths as the age of the mother increases. This aligns with existing research, suggesting that teenage mothers may lack awareness of maternal care, potentially contributing to higher child mortality rates [24]. Financial status proves to be a crucial factor, as mothers from affluent families exhibit a lower number of child deaths compared to their economically disadvantaged counterparts. This finding resonates with prior studies and emphasizes the intricate link between financial resources and the availability of healthcare services for mothers and children [25,26]. The study underscores the challenges faced by impoverished parents, who may struggle to afford even basic needs, let alone essential healthcare services. Additionally, the study sheds light on the impact of birth status, revealing a strong association between multiple births and increased child mortality rates. This observation aligns with conclusions drawn from similar investigations [27,28]. The challenges associated with managing the health and well-being of multiple children simultaneously likely contribute to this phenomenon. Furthermore, regional disparities emerge as a notable factor influencing child mortality rates. The result aligns with previous research by identifying variations in the number of child deaths per mother across different regions [29,30]. This regional dimension emphasizes the importance of tailoring healthcare policies and interventions to address specific challenges faced by communities in different geographical areas.

## Conclusion

This study focuses on the driving role of parental education in reducing child mortality in Bangladesh. The analysis using zero-inflated negative binomial regression model identifies that higher levels of maternal and paternal education are associated with reduced child mortality, whereas multiple births and economic disadvantage increase the risk. In addition to these factors, other potential determinants have been identified from the model such as

maternal age at first birth, birth status, and regional differences. We suggest the policy makers in Bangladesh to enhance parental education, along with addressing socioeconomic disparities and regional variations to improve child health outcomes. Prioritizing these interventions within national health strategies could significantly contribute to reducing child mortality and promoting child well-being in Bangladesh.

## Acknowledgment

We express our appreciation to the DHS (Demographic and Health Surveys) for granting us access to their dataset for our research. Furthermore, we acknowledge the National Institute of Population Research and Training (NIPORT) for conducting the BDHS, 2017–2018.

## Author contributions

**Conceptualization:** Farzana Afroz, Md. Muddasir Hossain Akib, Abida Sultana Asha.

**Data curation:** Abida Sultana Asha.

**Formal analysis:** Farzana Afroz.

**Methodology:** Farzana Afroz.

**Visualization:** Bikash Pal.

**Writing – original draft:** Farzana Afroz, Md. Muddasir Hossain Akib, Bikash Pal.

**Writing – review & editing:** Farzana Afroz, Md. Muddasir Hossain Akib, Bikash Pal, Abida Sultana Asha.

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
