## [Decision Letter · Decision Letter 0]

22 Oct 2024

PONE-D-24-22195Impact of parental education on number of under five children death per mother in BangladeshPLOS ONE

Dear Dr. Pal,

Thank you for submitting your manuscript to PLOS ONE. After careful consideration, we feel that it has merit but does not fully meet PLOS ONE’s publication criteria as it currently stands. Therefore, we invite you to submit a revised version of the manuscript that addresses the points raised during the review process. Please submit your revised manuscript by Dec 06 2024 11:59PM. If you will need more time than this to complete your revisions, please reply to this message or contact the journal office at plosone@plos.org . Please include the following items when submitting your revised manuscript:

We look forward to receiving your revised manuscript.

Kind regards,

Mohamed R. Abonazel, Ph.D.

Academic Editor

PLOS ONE

Journal Requirements: When submitting your revision, we need you to address these additional requirements. 1. Please ensure that your manuscript meets PLOS ONE's style requirements, including those for file naming. The PLOS ONE style templates can be found at https://journals.plos.org/plosone/s/file?id=wjVg/PLOSOne_formatting_sample_main_body.pdf and https://journals.plos.org/plosone/s/file?id=ba62/PLOSOne_formatting_sample_title_authors_affiliations.pdf 

**Additional Editor Comments:**

One reviewer suggested adding some references; please check these references and then add only those references related to the topic of the manuscript or the statistical analysis used in the applied study.

Reviewers' comments:

Reviewer's Responses to Questions

**Comments to the Author**

1. Is the manuscript technically sound, and do the data support the conclusions?

Reviewer #1: Yes

Reviewer #2: Yes

2. Has the statistical analysis been performed appropriately and rigorously?

Reviewer #1: Yes

Reviewer #2: Yes

3. Have the authors made all data underlying the findings in their manuscript fully available?

Reviewer #1: Yes

Reviewer #2: Yes

4. Is the manuscript presented in an intelligible fashion and written in standard English?

Reviewer #1: Yes

Reviewer #2: Yes

5. Review Comments to the Author

Reviewer #1: The manuscript presents the potential influence of the parental education and some sociodemographic parameters on child mortality in Bangladesh using the BDHS survey 2017-18. The manuscript could add value to the literature and inform the potential health interventions to accelerate the child mortality reduction in Bangladesh.

The methods section needs improvement. The results section can be further improved. The specific comments are mentioned in the attached document.

Reviewer #2: 1. The "Conclusion" section needs improvement.

2. The paper contains a few grammatical errors. The authors need to review the full text carefully.

3. Recently published papers are relevant to this manuscript, so one or all of them should be cited, such as

doi: https://doi.org/10.1080/00036846.2024.2313594

doi: 10.1017/S0954579424000725

4. In some equations, some symbols are not defined. Please correct this.

5. Add more statistical details about the Zero-Inflated Poisson and Zero-Inflated negative binomial models. See https://scik.org/index.php/cmbn/article/view/5658.

6. PLOS authors have the option to publish the peer review history of their article (what does this mean? ). If published, this will include your full peer review and any attached files.

**Do you want your identity to be public for this peer review?** For information about this choice, including consent withdrawal, please see our Privacy Policy .

Reviewer #1: **Yes: ** MANOJA KUMAR DAS

Reviewer #2: No

---

## [Author Response · Author response to Decision Letter 1]

6 Dec 2024

Authors’ responses to the points raised by the Academic Editor and reviewers

First of all, we would like to thank the Academic Editor and the reviewers for reviewing our joint work entitled “Impact of Parental Education on Number of Under Five Children Death Per Mother in Bangladesh” and providing their valuable suggestions about our manuscript. All the suggestions raised by the Academic Editor and reviewers have been addressed in the revised version which are categorically pointed out in the followings.

Responses to Reviewer 1:

1. We appreciate the constructive comment regarding the further improvement of the ‘Methods’ and ‘Results’ sections. In response, we have made several improvements to both sections, which are marked in the ‘Revised Manuscript with Track Changes.’

2. We are thankful for the comment suggesting that we should specify in the abstract whether our study utilizes a primary or secondary dataset. In response, we have incorporated this information into the abstract that our study utilizes a secondary dataset, specifically extracted from the Bangladesh Demographic and Health Survey (BDHS), 2017-18.

3. Based on the suggestion provided, we have added a line (line number 90) at the end of the ‘Introduction’ section to clarify that our study utilizes national survey data from the Bangladesh Demographic and Health Survey (BDHS), 2017-18.

4. Regarding the comment on the involvement of the Institutional Review Board (IRB) of ICF International, Rockville, Maryland, USA, in the ‘Ethics approval and consent to participate’ section, we would like to clarify that the IRB of ICF International was responsible for reviewing and approving the ethical aspects of the data collection process for the 2017–18 Bangladesh Demographic and Health Survey (BDHS). This approval ensured that the survey adhered to ethical guidelines concerning participant consent, confidentiality, and data handling. While ICF International played a key role in overseeing the ethical approval of the entire DHS Program, including the 2017–18 Bangladesh DHS, it was not directly involved in the day-to-day data collection in Bangladesh. The data collected through BDHS was then made available for secondary analysis, which is the dataset we have utilized in our study.

5. We appreciate the comment on the ‘Statistical analysis’ subsection to present the overall approach for the analysis in terms of population distribution, divisions, and others. We would like to clarify that the primary focus of this study is to evaluate the impact of parental education on under-five child mortality across the entire population of Bangladesh, rather than analyzing variations across population distributions or subgroups. To maintain this focus, we prioritized regression modeling as the core analytical approach, specifically utilizing Poisson, zero-inflated Poisson, and zero-inflated negative binomial models. These models allowed us to assess the effects of parental education while controlling for other covariates such as administrative division, place of residence, wealth index, and birth status.

6. We thank the reviewer for the observation regarding the need to specify the types of residence considered in the ‘Results’ section. In response, we have clarified this in the revised text by explicitly stating that urban or rural residency was analyzed and found to be insignificant.

7. Regarding the comment on Socioeconomic Status (SES), it is typically characterized in the DHS dataset and divided based on a wealth index which was originally created through principal component analysis (PCA) using household asset data. This index is based on household asset ownership, type of housing, access to electricity, sanitation, and other living standard indicators. In BDHS 2017-18 dataset, the wealth index was initially categorized into five groups: poorest, poorer, middle, richer, and richest. In this study, we further combined these categories into three groups: poor (poorest and poorer), middle, and rich (richer and richest).

8. In response to the reviewer’s comment regarding the variable ‘Birth status’ in the ‘Results’ section, we would like to clarify that the variable is categorized as follows: mothers who have given birth to more than two children are categorized as ‘Multiple’, while those who have given birth to one or two children are categorized as ‘Single/Double’. This classification is explained at the last line in the ‘Data and Variables’ section of the manuscript.

9. Based on the comment on Table 2 heading, we would like to clarify that the input variables for the regression models are described in detail in the ‘Data and Variables’ subsection of the ‘Methods’ section. In this subsection, we have clearly outlined the response variable (number of under-five children deaths per mother) as well as the explanatory variables such as maternal and paternal education, wealth index, birth status, administrative division, place of residence, exposure to media, and mother's working status. We believe this section adequately addresses the variables used in the models, and we hope this clears up any confusion.

10. Regarding the ‘Region’ variable, we chose Dhaka as the comparator category because it is the capital of Bangladesh and is generally more economically and socially enriched compared to other divisions. By selecting Dhaka as the baseline, we aimed to compare the other divisions/regions against a more developed and resource-rich area.

Responses to Reviewer 2:

1. We appreciate the constructive comment regarding the improvement of the ‘Conclusion’ section. In response, several revisions have been made to this section, which are highlighted in the ‘Revised Manuscript with Track Changes.’

2. The comment regarding grammatical errors is acknowledged. The full text has been carefully reviewed, and necessary corrections have been made.

3. We appreciate the suggestion to cite recently published papers. However, after reviewing the two papers mentioned, the authors find that they do not align closely with the focus of our study. These papers primarily address effective parenting, whereas our study specifically explores the relationship between parental education and under-five mortality. Since parenting and parental education are distinct concepts, we have chosen not to cite these papers in our work.

4. The issue regarding undefined symbols in some equations has been addressed. All symbols have been appropriately defined, and the corrections are marked in the ‘Revised Manuscript with Track Changes.’

5. Additional statistical details about the Zero-Inflated Poisson and Zero-Inflated Negative Binomial models have been included as suggested. The revisions are clearly marked in the ‘Revised Manuscript with Track Changes.’

---

## [Decision Letter · Decision Letter 1]

22 Jan 2025

Impact of parental education on number of under five children death per mother in Bangladesh

PONE-D-24-22195R1

Dear Dr. Pal,

We’re pleased to inform you that your manuscript has been judged scientifically suitable for publication and will be formally accepted for publication once it meets all outstanding technical requirements.

Kind regards,

Mohamed R. Abonazel, Ph.D.

Academic Editor

PLOS ONE

Reviewers' comments:

Reviewer's Responses to Questions

**Comments to the Author**

1. If the authors have adequately addressed your comments raised in a previous round of review and you feel that this manuscript is now acceptable for publication, you may indicate that here to bypass the “Comments to the Author” section, enter your conflict of interest statement in the “Confidential to Editor” section, and submit your "Accept" recommendation.

Reviewer #2: All comments have been addressed

2. Is the manuscript technically sound, and do the data support the conclusions?

Reviewer #2: Yes

3. Has the statistical analysis been performed appropriately and rigorously?

Reviewer #2: Yes

4. Have the authors made all data underlying the findings in their manuscript fully available?

Reviewer #2: Yes

5. Is the manuscript presented in an intelligible fashion and written in standard English?

Reviewer #2: Yes

6. Review Comments to the Author

Reviewer #2: The authors have adequately addressed the comments, and I feel this manuscript is now acceptable for publication.

7. PLOS authors have the option to publish the peer review history of their article (what does this mean? ). If published, this will include your full peer review and any attached files.

**Do you want your identity to be public for this peer review?** For information about this choice, including consent withdrawal, please see our Privacy Policy .

Reviewer #2: No

---

## [Editor Report · Acceptance letter]

PONE-D-24-22195R1

PLOS ONE

Dear Dr. Pal,

I'm pleased to inform you that your manuscript has been deemed suitable for publication in PLOS ONE. Congratulations! Your manuscript is now being handed over to our production team.

Kind regards,

on behalf of

Dr Mohamed R. Abonazel

Academic Editor

PLOS ONE